# Human Activity Classification Based on Angle Variance Analysis Utilizing the Poincare Plot

Solaiman Ahmed [1],* , Tanveer Ahmed Bhuiyan [2], Taiki Kishi [1], Manabu Nii [1] and Syoji Kobashi [1]

[1]  Graduate School of Engineering, University of Hyogo, Kobe 671-2280, Japan;
ei19j006@steng.u-hyogo.ac.jp (T.K.); nii@eng.u-hyogo.ac.jp (M.N.); kobashi@eng.u-hyogo.ac.jp (S.K.)
[2]  Demant A/S, Kongebakken 9, 2765 Smørum, Denmark; tabh@demant.com
*   Correspondence: el19r001@steng.u-hyogo.ac.jp

**Featured Application: This research work aims to provide a sensor-based body angle variance model to classify static and dynamic activities along with their variants, which has an impact on being healthy.**

**Abstract:** We propose a single sensor-based activity classification method where the Poincare plot was introduced to analyze the variance of the angle between acceleration vector with gravity calculated from the raw accelerometer data for human activity classification. Two datasets named 'Human Activity Recognition' and 'MHealth dataset' were used to develop the model to classify activity from low to vigorous intensity activities and posture estimation. Short-term and long-term variability analyzing the property of the Poincare plot was used to classify activities according to the vibrational intensity of body movement. Commercially available Actigraph's activity classification metric 'count' resembled value was used to compare the feasibility of the proposed classification algorithm. In the case of the HAR dataset, laying, sitting, standing, and walking activities were classified. Poincare plot parameters SD1, SD2, and SDRR of angle in the case of angle variance analysis and the mean count of X-, Y-, and Z-axis were fitted to a support vector machine (SVM) classifier individually and jointly. The variance- and count-based methods have 100% accuracy in the static–dynamic classification. Laying activity classification has 100% accuracy from other static conditions in the proposed method, whereas the count-based method has 98.08% accuracy with 10-fold cross-validation. In the sitting–standing classification, the proposed angle-based algorithm shows 88% accuracy, whereas the count-based approach has 58% accuracy with a support vector machine classifier with 10-fold cross-validation. In the classification of the variants of dynamic activities with the MHealth dataset, the accuracy for angle variance-based and count-based methods is 100%, in both cases, for fivefold cross validation with SVM classifiers.

**Keywords:** Poincare plot; count; tilt angle; body posture; vibrational intensity

## 1. Introduction

Any bodily movement produced by skeletal muscles increasing energy expenditure above a basal level is called physical activity (PA) [1]. It can be divided into two main categories—exercise and non-exercise PA. The intensity level can classify both as light, moderate, and vigorous. Separation of human activities based on the intensity level is defined as activity classification [2,3].

Activity classification can be vision- or sensor-based. There are many constraints for vision-based activity classification, primarily because of privacy and installation cost issues [4].

From the low-cost and continuous health monitoring perspective, sensor-based human activity classification has become very popular [5]. For estimating stress parameters in the case of ambulatory and static conditions, sensor-based activity monitoring is a

commonly used solution to synchronize physiological data (e.g., ECG and PPG) with activity monitoring sensor data [6]. Continuous assessment of stress can avoid unexpected accidents, especially for the workers' life in many heavy industries. Understanding the normal and abnormal ambulatory physiological condition is dependent on the prior and present status of PA [7–9].

Many deep learning-based methods, which are complex systems for wearable devices, have been proposed for human activity classification, especially with the University of California, Irvine Human Activity Recognition dataset [10,11]. As deep learning has automatic feature extraction capabilities, those methods do not require any manual feature extraction techniques. In most cases, a 1D convolutional neural network and temporal sequence-based long–short term memory neural network had been used for feature extraction and classification for human activity [12–14], despite having some constraints for wearable devices in terms of computational complexity [15]. There are many studies using the conventional machine learning approach where fast feature extraction methods have been proposed to optimize an algorithm with multiple features [16].

Along with machine learning development, many industries have grown worldwide to develop hardware [17] and software products to monitor human activity, and Actigraph is one of them. It is a leading provider of wearable physical activity and sleep monitoring solutions for the global scientific community [18]. Actigraph's research-grade accelerometry monitors are the most widely used and extensively validated devices of their kind [19]. Its monitoring solutions have been used in thousands of academic studies worldwide to objectively measure the physical activity, mobility, and sedentary behavior of study participants in real-world and laboratory settings. A flexible and robust software technology platform provides the most comprehensive data monitoring, analysis, and management support in the industry [20–22].

Actigraph calculates count values based on activity intensity, and the hip is the best location to calculate the intensity of PA, but its ability to detect body posture is inadequate in a single worn accelerometer [23]. The commonly used inclinometer measures the angle between the gravity direction and the acceleration vector, but it can detect the posture only with 70% accuracy at best [24,25]. In the case of thigh-worn accelerometers, which measure the thigh inclination angle, it can differentiate between sitting and standing with over 90% accuracy. However, differentiation of sitting from laying is difficult because of different orientations for both postures [26–28]. A combination of the angle of rotation and inclination for thigh has shown promising results to differentiate sitting from laying [29]. Multiple accelerometer-based solutions have reached over 98% accuracy in identifying body posture [30,31]. However, in the case of a large-scale population, less complex (single-worn) and high feasibility systems are of primary importance besides the ability to provide information on light intensity behavior to the researchers. Obviously, one has to face a trade-off between simple systems and more burdensome and complex systems offering higher accuracy [32].

Variance is an important parameter for classifying activities. Poincare plot-based activity classification is an investigation aimed to develop a simple variance-based classification method that may be less complex in terms of computational complexity than a deep learning-based system. To the best of our knowledge, there is no research on variance-based activity classification, so this paper was warranted for such investigation. Poincare plot can be a powerful tool for analyzing the variability of the angle derived from a single waist- and chest-mounted accelerometer. A 2D plot can be constructed by plotting consecutive points of angle in time intervals into series on a Cartesian plane. It is usually used for qualitative visualization of heart rate (HR) interval because of its ability to measure the short-term and long-term HR variability. In the determination of physical activity intensity, low (like short-term) and high (like long-term) angle variability analyzing property can be useful, the same as the Actigraph 'count' value, to classify activities.

In this paper, we propose a single sensor-based activity classification method where Poincare plots have been introduced to analyze the variance of angle calculated from the raw accelerometer data.

## 2. Materials and Methods

### 2.1. Dataset

Two datasets were used in this research. The first one is the Human Activity Recognition dataset, and the second one is the MHealth dataset from the UCI data repository [33,34]. In the Human Activity Recognition dataset, light to moderate intensity activities are available (e.g., laying to walking) and vigorous activity like running is not available. Although the MHealth dataset has different data acquisition protocol, it contains vigorous activity (e.g., running). Thus, MHEALTH data were also used in this research.

The experiments of the Human Activity Recognition (HAR) dataset collection were carried out with a group of 30 volunteers within an age bracket of 19–48 years. Each person performed six activities (walking, walking upstairs, walking downstairs, sitting, standing, and laying) while wearing a smartphone (Samsung Galaxy S II) on the waist. Three-axial linear acceleration and three-axial angular velocity were logged at a constant rate of 50 Hz using its embedded accelerometer and gyroscope.

Laying, sitting, and standing data were considered as static activities, and walking, walking up, and walking down were considered as dynamic activities. In the case of the HAR dataset, the first 20 s of data were used in this experiment, because some unexplained jump that may be induced by hardware default has been explained in the limitation section.

In the MHealth dataset, body motion and vital signs were recorded for ten volunteers of diverse profiles while performing several physical activities (e.g., walking, running, and so on). Sensors placed on the subject's chest, right wrist, and left ankle are used to measure the motion experienced by diverse body parts. The sensor positioned on the chest also provides two-lead ECG measurements, which can potentially be used for basic heart monitoring and examining the effects of exercise from the ECG.

Although the MHealth dataset contain three accelerometers in different body positions, among them, the chest is the only location that is more stable with respect to the waist compared with the wrist and ankle. As the HAR dataset was acquired from a waist-mounted smartphone, to validate the different datasets with the same model, data coherence was an importance consideration. So, only the chest-mounted accelerometer named Shimmer 2 [BUR10] sensor with constant rate of 50 Hz was taken into consideration for the MHealth dataset.

### 2.2. Proposed Methods

A three-axis accelerometer provides a vector that comprises components of gravity on the local coordinate frame. In the case of the HAR dataset, activity was recorded with an accelerometer in a smartphone (Samsung Galaxy S II). As the smartphone had different initial orientations from person to person for data estimation, normalizing of all the measurements was carried out by keeping the laying data as the reference orientation. This was accomplished by calculating the initial orientation of the z-axis of the accelerometer frame with respect to the gravity vector. After initializing the orientation of the device, the angle (degree) was derived between the gravity and accelerometer vector for every sample. Poincare plot parameters SD1, SD2, and SDRR were calculated for angle values, and parameters were fitted to a support vector machine classifier.

Multi-classification was implemented in three steps. The step by step classification process is shown in Figure 1.

At first, the classification between static and dynamic activities was done. After that, variants of static and dynamic activities were classified with mean angle and variance analysis. The Actigraph count-based activity classification method was studied for all cases, and all the analyses were accomplished with 10-fold cross-validation with an SVM classifier. In sedentary body posture estimation, more specifically sitting and standing

classification, count-based classification is not satisfactory, where an angle-based method can be a better solution.

A comparative study between angle- and count-based activity classification and modification (sitting and standing classification) with the angle-based over count-based method were shown with novel angle variance analysis for human activity recognition.

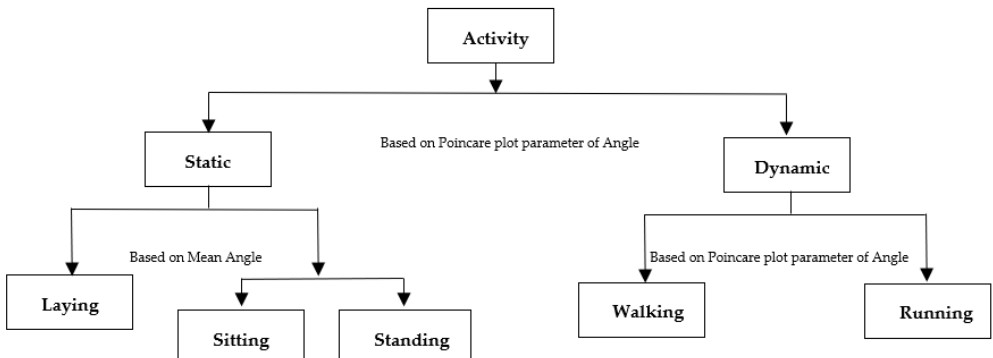

**Figure 1.** Step by step activity classification process.

### 2.3. Orientation of the Sensor and Angle Calculation with Gravity

The rotation matrix was derived from the laying data of each subject applied to other conditions (sitting and standing) to have a common reference (Figure 2). It was calculated as shown in Manon et al. [35].

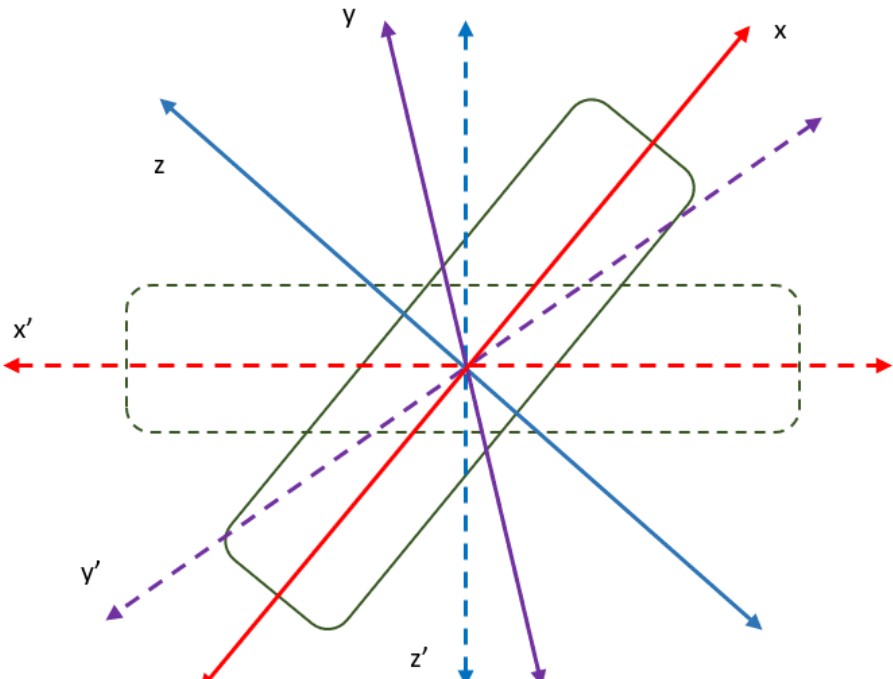

**Figure 2.** The orientation of the laying data (dotted line) after the multiplication with rotation matrix from unoriented position (solid line).

### 2.4. Poincare Plot Explanation

Poincare plot is a geometrical representation of a time series in a Cartesian plane. Heart rate dynamic is a nonlinear process, and its pattern can be explained by Poincare plot [36]. By plotting consecutive points of R-peak (QRS) to R-peak time interval into series on a Cartesian plane, a 2D plot was constructed [37]. It is extensively used for qualitative visualization of the physiological signal and commonly applied to assess the dynamics of heart rate variability (HRV) [36–38].

An ellipse can be fitted to the shape of the Poincare plot. SD1 and SD2, along with semi-minor and semi-major axis, respectively, are used for quantification of the Poincare plot geometry.

The description of SD1 and SD2 in terms of linear statistics shows that the standard descriptors guide the visual inspection of the distribution. In terms of HRV, it reveals a useful visual pattern of the RR interval data by representing both short- and long-term variations of the signal [39,40].

$$SD1 = \sqrt{\frac{\left(\frac{1}{L}\right) \sum ((x_t - x_{t-1}) - \overline{(x_t - x_{t-1})})^2}{2}} \tag{1}$$

$$SD2 = \sqrt{\frac{\left(\frac{1}{L}\right) \sum ((x_t + x_{t-1}) - \overline{(x_t + x_{t-1})})^2}{2}} \tag{2}$$

$$SDRR = \sqrt{\frac{\left(\sqrt{\frac{\left(\frac{1}{L}\right) \sum ((x_t - x_{t-1}) - \overline{(x_t - x_{t-1})})^2}{2}}\right)^2 + \left(\sqrt{\frac{\left(\frac{1}{L}\right) \sum ((x_t + x_{t-1}) - \overline{(x_t + x_{t-1})})^2}{2}}\right)^2}{2}} \tag{3}$$

We used this concept to determine the non-linearity of angle series for human activity classification purposes with an accelerometer sensor. Poincare plot derived from an angle can also be effective for the classification of variations of dynamic activities like walking and running.

### 2.5. Feature Extraction

From raw accelerometer data, the angle between the acceleration vector and gravity vector was calculated. The mean angle and Poincare plot parameters of angle were calculated from the angle. These parameters were fitted to the support vector machine classifier to classify human activity. Static and dynamic activities were classified on the basis of the Poincare plot parameter. Classification between static activities was classified on the basis of mean angle and dynamic activities were classified on the basis of Poincare plot parameters. The process for feature extraction is shown in Figure 3.

### 2.6. Actigraph Count Calculation Procedure

Several researchers have tried to reproduce the working algorithm of the Actigraph. The validity of the Actigraph for the assessment of physical activity in a population has been extensively studied across many different ages, groups, gender, and patients [41].

According to Brønd et al., in the calculation of the value that resembles the Actigraph, the count can be generated from raw accelerometer data with the following steps. At first, the raw data were resampled to 30 Hz, and then they were band-passed with a [0.1–5] Hz filter. The band-passed signal was down sampled to 10 Hz. Saturation and dead band were applied to the data such that no acceleration exceeds the maximum absolute value (truncation threshold), and any value lower than the minimum value (dead band) is set to 0, respectively, to avoid any non-human activity. Lastly, the data were divided according to the time window and the summation of the values in each time window is the final activity count [41]. The Actigraph count calculation process is shown in Figure 4.

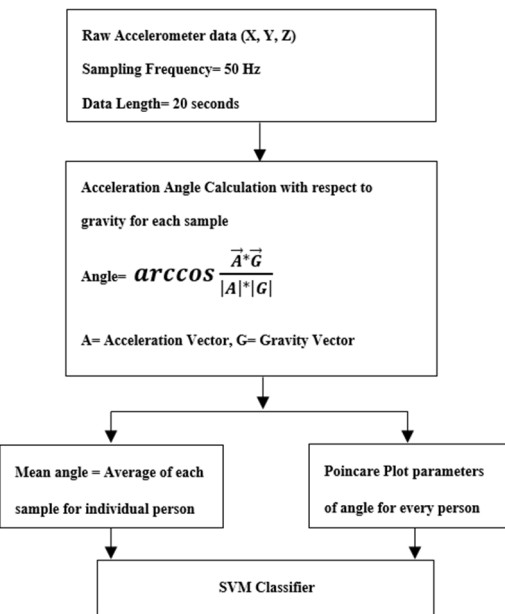

**Figure 3.** Feature extraction process.

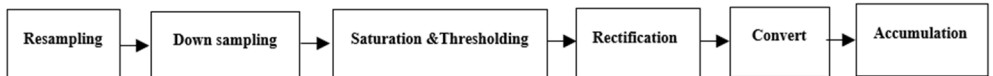

**Figure 4.** Actigraph count calculation process.

## 3. Results

### 3.1. Static and Dynamic Activities' Classification

Static and dynamic activities were classified from the HAR dataset, where laying, sitting, and standing were considered static activities. Furthermore, walking, walking up, and walking down were considered dynamic activities. The angle derived from raw accelerometer data for all activities and the corresponding Poincare plot is shown in Figures 5 and 6.

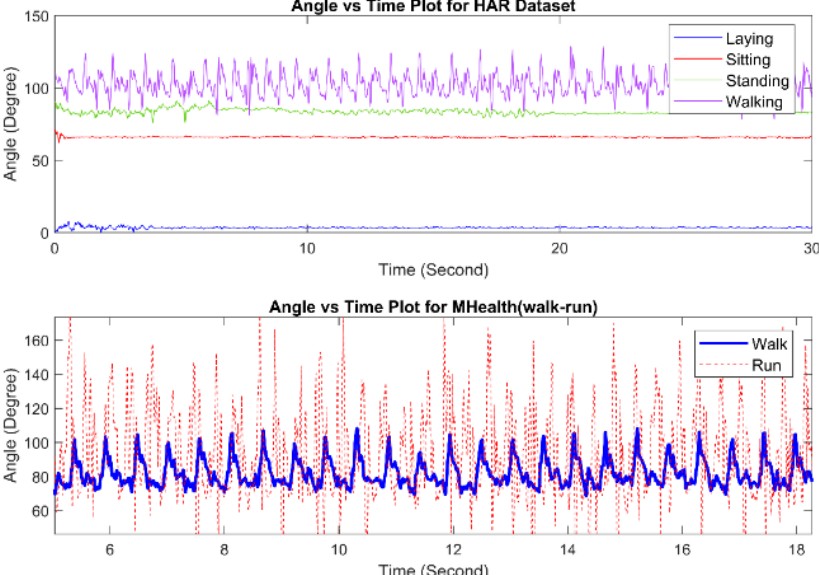

**Figure 5.** Plots of angle between acceleration vector and gravity. The up panel denotes the laying, sitting, standing, and walking angle for the Human Activity Recognition (HAR) dataset and the down panel walking and running angle for the MHealth dataset.

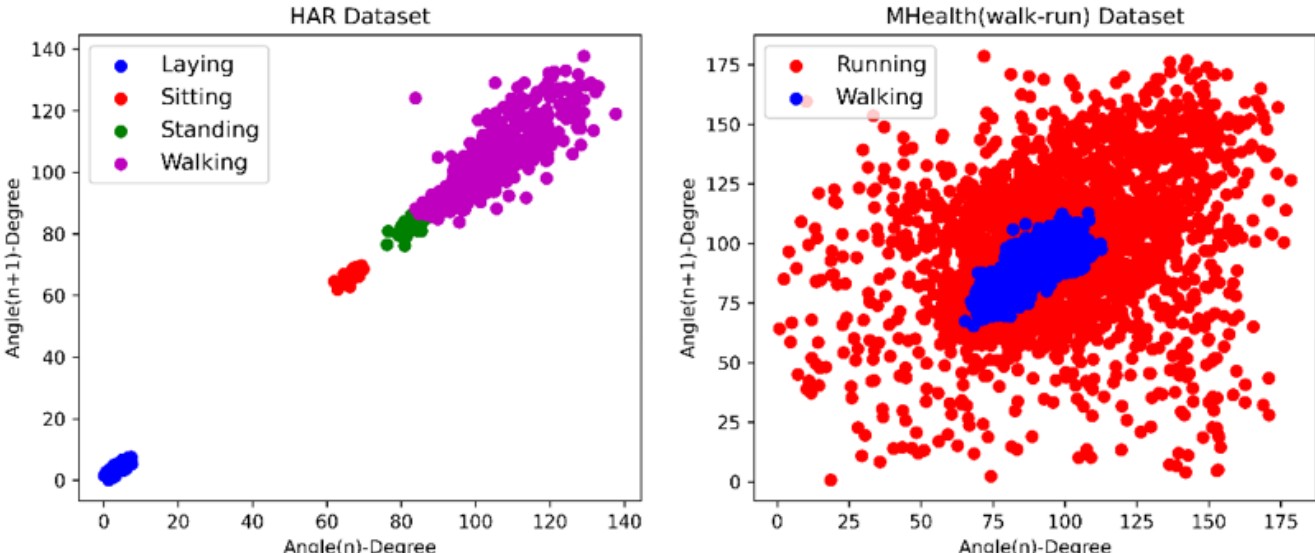

**Figure 6.** Poincare plot of angle. The left panel denotes laying, sitting, standing, and walking Poincare plot of angle for the HAR dataset and the right panel walking and running Poincare plot of angle for MHealth.

Poincare plot parameters SD1, SD2, and SDRR were calculated and classified with an SVM classifier with linear kernel, where C = 1 and gamma = 'scale'. The mean and standard deviation for all activities are shown in Figures 7 and 8. The Actigraph count resembles that the value was analyzed where count calculation for static and dynamic activities was considered.

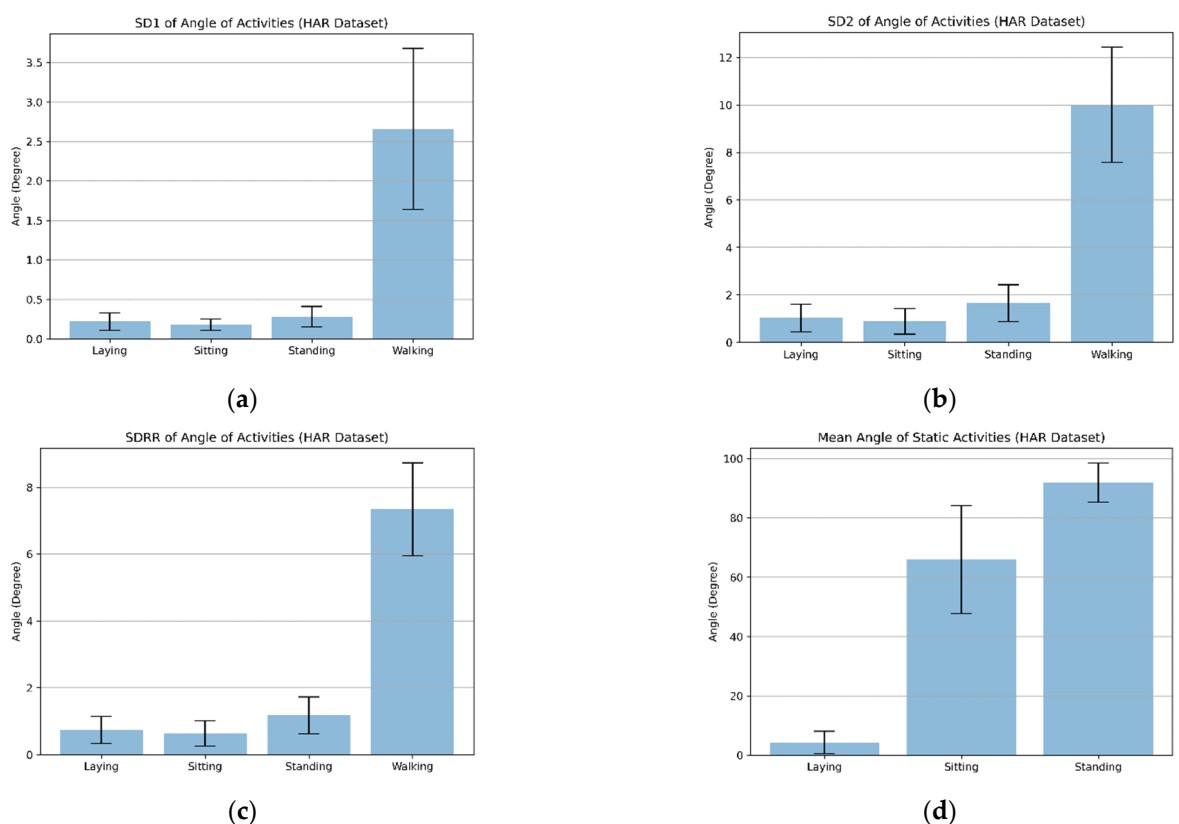

**Figure 7.** (**a**–**c**) Bar plot of SD1, SD2, and SDRR, respectively, for laying, sitting, standing, and walking. (**d**) Bar plot of the mean angle for static activities with the HAR dataset.

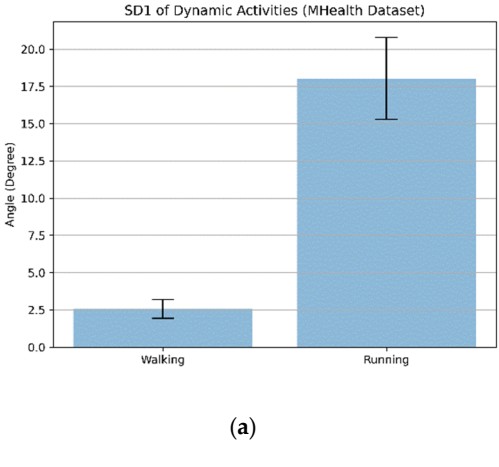

(**a**)

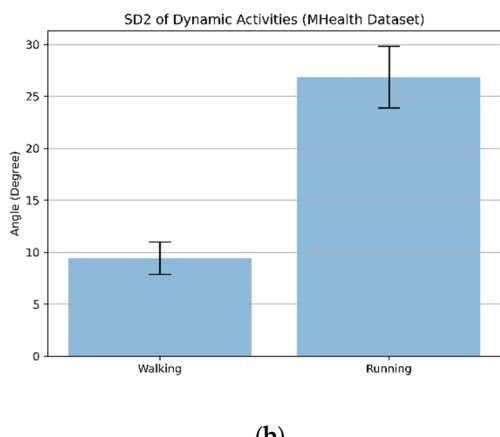

(**b**)

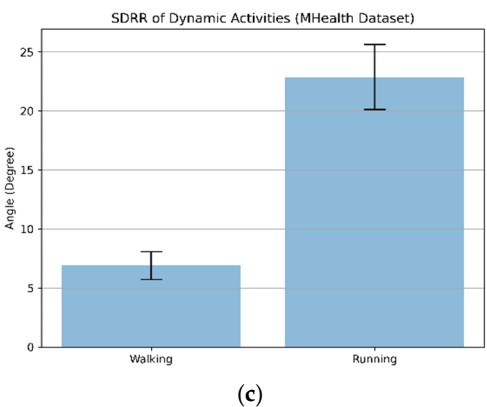

(**c**)

**Figure 8.** (**a**–**c**) Bar plot of SD1, SD2, and SDRR, respectively, of dynamic activities (walking and running) with the MHealth dataset.

Table 1 shows the result for static and dynamic activities for the angle- and count-based classification system with accuracy (100%), sensitivity (100%), and specificity (100%). The ROC curve for static and dynamic classification with the Actigraph count and Poincare plot-based classification is shown in Figure 9.

In the case of angle-based classification, SD1, SD2, and SDRR were considered separately and together as a feature to classify. In contrast, the mean counts of X, Y, and Z were considered separately and together.

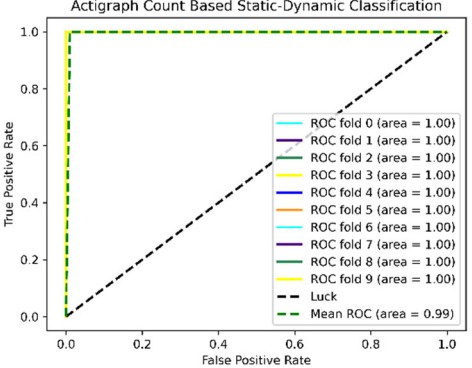

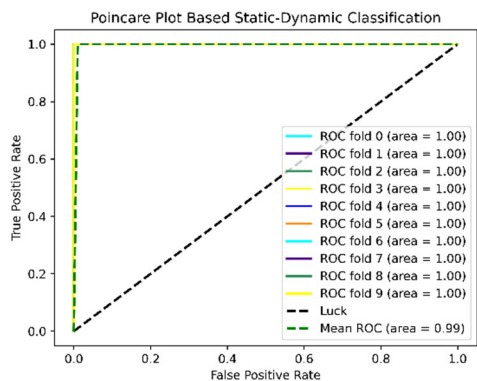

**Figure 9.** ROC curve for static and dynamic classification. The left panel denotes the ROC curve for the Actigraph count-based static and dynamic activities' classification. The right panel denotes the Poincare plot-based static and dynamic activities' classification.

**Table 1.** Accuracy with count-based and angle-based classification for the HAR dataset.

| | | Accuracy (%) | Sensitivity (%) | Specificity (%) |
|---|---|---|---|---|
| **Static/Dynamic Classification** | | | | |
| Actigraph count | X | 100 | 100 | 100 |
| | Y | 99 | 98 | 100 |
| | Z | 98 | 98 | 99 |
| | X, Y, Z | 100 | 100 | 100 |
| Poincare plot parameters | SD1 | 100 | 100 | 100 |
| | SD2 | 100 | 100 | 100 |
| | SDRR | 100 | 100 | 100 |
| | SD1, SD2, SDRR | 100 | 100 | 100 |
| **Laying–Sitting/Standing Classification** | | | | |
| Actigraph count | X | 98 | 98 | 98 |
| | Y | 71 | 34 | 74 |
| | Z | 69 | 66 | 69 |
| | X, Y, Z | 98 | 98 | 98 |
| Angle | Mean Angle | 100 | 100 | 100 |
| **Sitting-Standing Classification** | | | | |
| Actigraph count | X | 48 | 50 | 0 |
| | Y | 55 | 73 | 52 |
| | Z | 46 | 48 | 43 |
| | X, Y, Z | 57 | 67 | 54 |
| Angle | Mean Angle | 88 | 97 | 82 |

### 3.2. Laying and Sitting–Standing Classification

Laying and sitting–standing activities were classified with the mean angle from the static activities. After the initial orientation, where laying data were considered as a reference, it shows the best accuracy (100%), sensitivity (100%), and specificity (100%). Overall, the count-based classification result shows accuracy (98.26%), sensitivity (98.14%), and specificity (98.31%). All the results were analyzed with 10-fold cross-validation. The ROC curve for laying and other static classification with the Actigraph count and mean angle-based classification is shown in Figure 10.

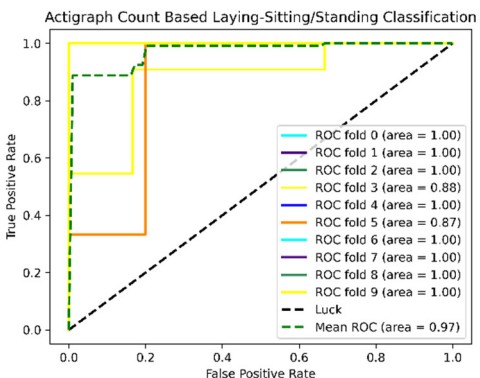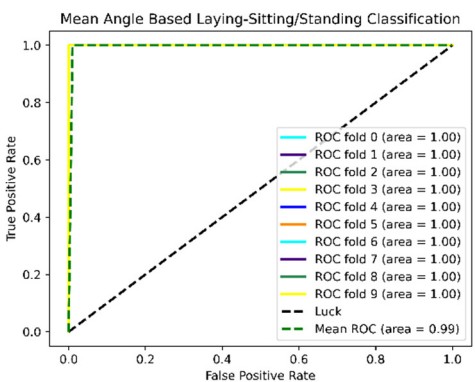

**Figure 10.** ROC curve for laying and other static classification. The left panel denotes the ROC curve for the Actigraph count-based laying and other static classification. The right panel denotes the mean angle-based static and dynamic activities' classification.

### 3.3. Sitting and Standing Classification

Sitting–standing activities were also classified with the mean angle after filtration of laying data from static activities. Mean angle-based classification shows accuracy (88.11%), sensitivity (97.36%), and specificity (82.53%). In the case of the count-based method, the mean count of X, Y, and Z shows accuracy (57.62%), sensitivity (67.85%), and specificity (54.44%) with 10-fold cross-validation. The ROC curve for sitting and standing classification with the Actigraph count and mean angle-based classification is shown in Figure 11.

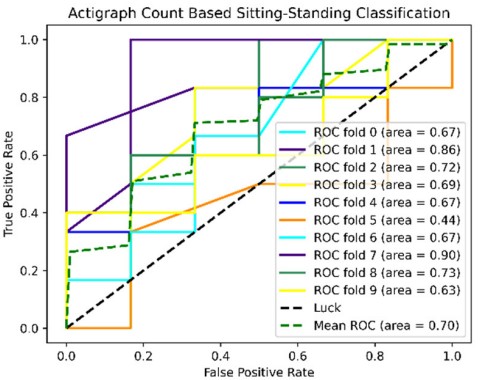 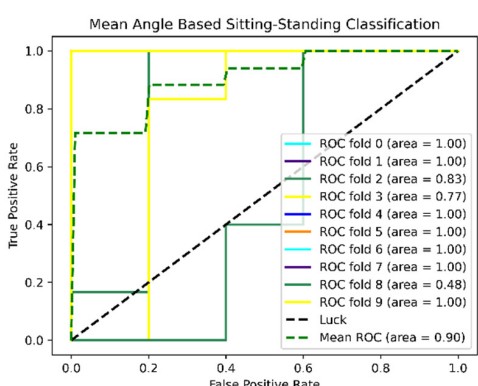

**Figure 11.** ROC curve for sitting and standing classification. The left panel denotes the ROC curve for the Actigraph count-based sitting and standing classification. The right panel denotes the mean angle-based static and dynamic activities' classification.

### 3.4. Walking and Running Classification

Angle variance analysis was considered as a medium (walk) to vigorous (run) activity classifier. Table 2 shows the classification between walking and running activities with count and angle variance analysis. The classification results are shown in Table 2 for SD1, SD2, and SDRR, as well as count of X, Y, and Z, in separate and combined form. The ROC curve for walking and running classification with the Actigraph count and Poincare plot-based classification is shown in Figure 12. The mean count of the X, Y, and Z-axis shows accuracy (100%) with sensitivity (100%) and specificity (100%), whereas SD1, SD2, and SDRR of angle show accuracy (100%) with sensitivity (100%) and specificity (100%).

**Table 2.** Accuracy with count- and angle variance (Poincare plot)-based classification for the MHealth (walk–run) dataset.

| Walking–Running Classification | | | | |
|---|---|---|---|---|
| | | Accuracy (%) | Sensitivity (%) | Specificity (%) |
| | X | 100 | 100 | 100 |
| | Y | 100 | 100 | 100 |
| Actigraph count | Z | 85 | 76 | 100 |
| | X, Y, Z | 100 | 100 | 100 |
| | SD1 | 100 | 100 | 100 |
| Poincare plot parameters | SD2 | 100 | 100 | 100 |
| | SDRR | 100 | 100 | 100 |
| | SD1, SD2, SDRR | 100 | 100 | 100 |

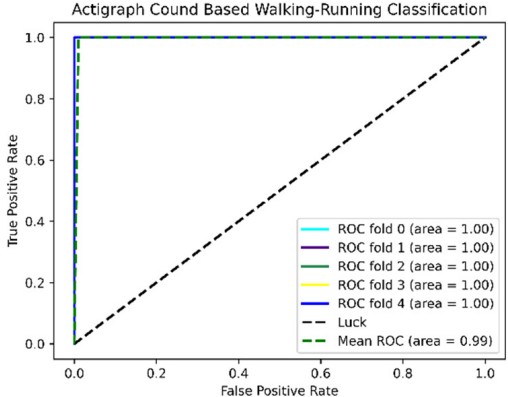 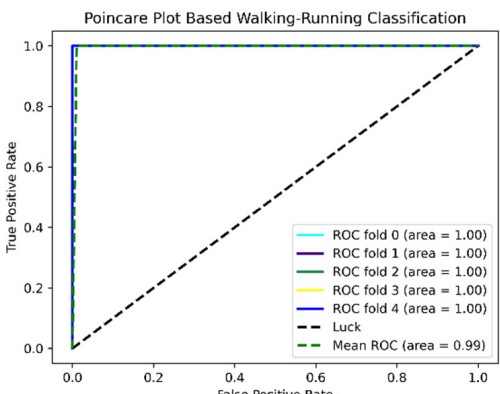

**Figure 12.** ROC curve for walking and running classification. The left panel denotes the ROC curve for the Actigraph count-based walking and running classification. The right panel denotes the Poincare plot-based walking and running activities' classification.

## 4. Discussion

In this study, a Poincare plot-based variance analysis was applied to angles derived from raw accelerometer data to classify static and dynamic activities along with their variants (e.g., laying, sitting, and standing for static activities).

In terms of classification between static and dynamic activities, both count- and angle-based classification is evidently accurate. The acceleration variation is reflected significantly when the user is in motion, which thereby increases the variability in accelerometer counts and the angle.

In the case of laying and sitting–standing classification, the results for the mean count of the X-axis show accuracy (98.26%), sensitivity (98.14%), and specificity (98.31%), whereas the combined feature for the mean count of three axes together shows the same results because of standard scaling or normalization. However, the other two axes (Y and Z) do not perform well individually compared with the mean count of the X-axis. However, the results were found without the initial orientation of the senor.

In the case of count-based calculation for static activities, many subjects' mean count was zero (as the count method calculates the value after minimum vibration). After the initial orientation of the device, the mean angle was found to be a good marker for laying and sitting–standing classification, which is very much obvious from Figure 6.

In the case of classification between sitting and standing, we hypothesize that standing would induce more postural variability in the accelerometer owing to the balance retraining. Following this hypothesis, the classification between sitting and standing with the angle variance-based method yielded an accuracy of 68.08% with SD1. Eventually, we realized that the mean angle performed better in accuracy (88.11%) with sensitivity (97.39%) and specificity (82.53%). In contrast, the count-based method was performed with accuracy (57.62%), sensitivity (67.85%), and specificity (54.44%).

In terms of vigorous activity classification with walk and run data from the MHealth dataset, count (X, Y, Z) and SD (SD1, SD2, SDRR) were considered individually and jointly for the walking–running classification. The overall count- and variance-based classification accuracy was 100% and 100%, respectively, with five-fold cross-validation for SVM classifiers.

In this paper, we have shown a robust classification of activity using variance features from a body-mounted accelerometer. In comparison with some deep learning architectures in [42–44], the overall accuracy was 92.71%, 92.13%, and 92.67% with the 1D convolutional neural network (CNN), CNN-LSTM, and bidirectional LSTM, respectively. In [45], Bayes classifier is used where the time, frequency, and spatial domain totals of 19 features are extracted. Principal component analysis is applied to reduce the dimension, whereas the overall accuracy for the proposed method is around 96% for all activities without any dimensionality reduction with SD1, SD2, and SDRR. To the best of our knowledge, this

is the first study of such kind. Activity classification based on the accelerometer-only feature will enable cheaper and robust applications in telemedicine applications, as well as continuous monitoring of stress signature on humans. Such a classification method will have manifold advantages ranging from health applications to continuous patient monitoring, tracking the stress markers of industry workers. Hence, human activity classification with significant accuracy is warranted.

### 5. Conclusions and Perspectives

Waist-worn accelerometers can classify activities below a certain intensity threshold (e.g., count) as static behavior, while it is difficult to separate different body postures (e.g., sitting–standing) from each other. As standing may confer some health benefits compared with sitting and laying, it needs to be classified and estimated.

The proposed Poincare plot-based angle variance analysis approach can be a substitute to the Actigraph count-based method (which calculates activity intensity) for both (1) the single waist-worn HAR dataset and (2) the chest-worn MHealth (walk-run) dataset with a tri-axial accelerometer.

In this paper, we showed that our proposed method has good-to-excellent performance in identifying different body postures of sedentary activities as well as moderate (walk) to vigorous (run) activities, which may be deemed less computationally complex and cheaper to implement in wearable devices for ambulatory activity classification platform.

### 6. Limitations

Some data from sitting were avoided in the activity classification calculation because of random noise, as shown in Figure 13. In the case of SD calculation, those data resulted in outliers.

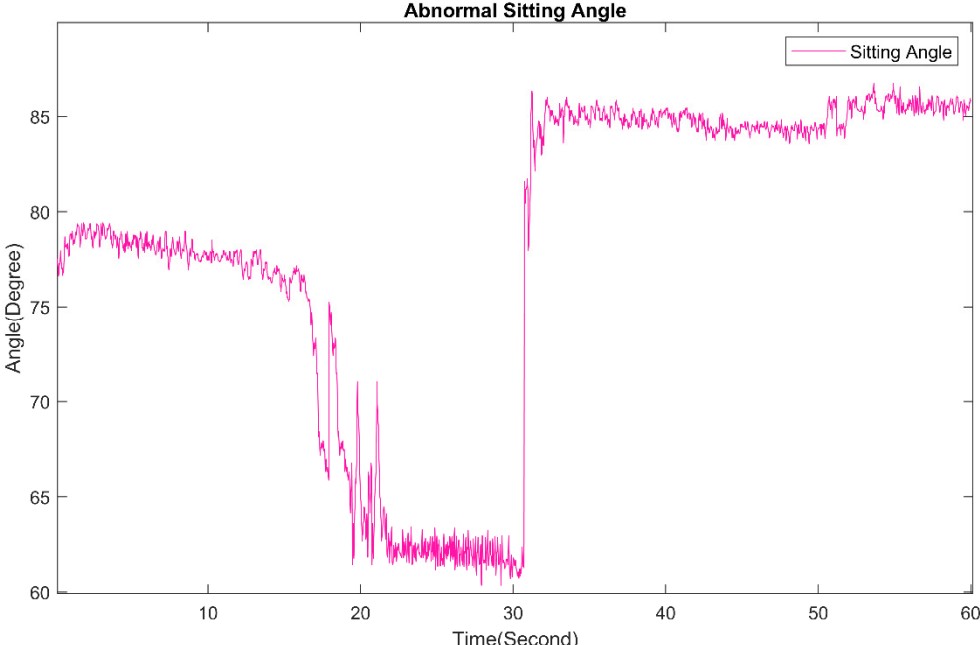

**Figure 13.** Plot of abnormal sitting angle.

In the case of sitting data from Figure 14, there is a jump in angle value that may be induced by a hardware fault. As a data cleaning approach, we avoided those unexplained jumps as human activity transition is unrealistic in 20 milliseconds (one sample).

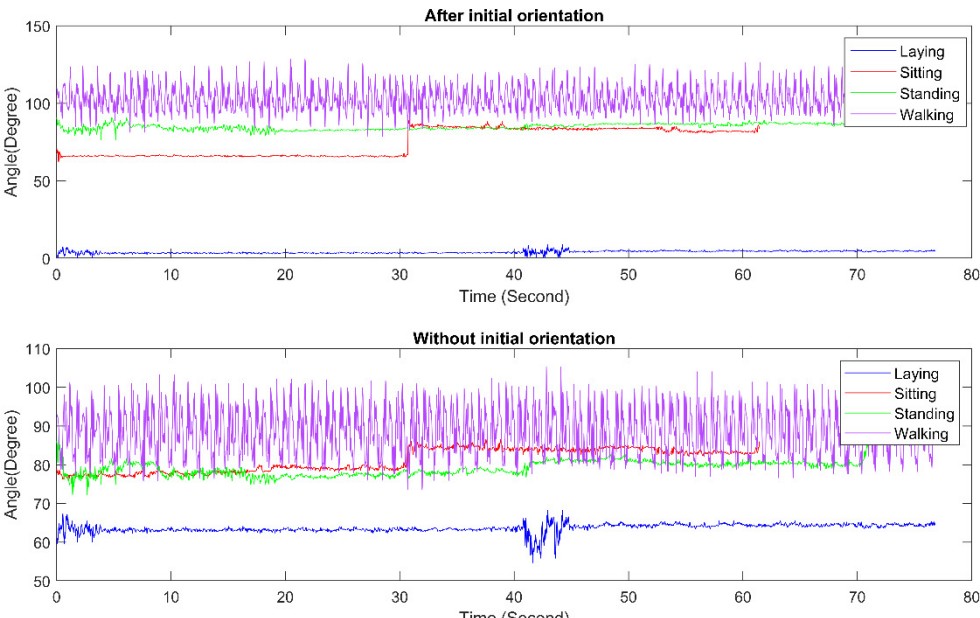

**Figure 14.** Plots of angle with a sudden jump in sitting angle. The up panel denotes angle of laying, sitting, standing, and walking for the HAR dataset after initial orientation and the down panel denotes angle without initial orientation.

**Author Contributions:** Conceptualization, S.A.; methodology, S.A. and T.A.B.; software, S.A.; validation, S.A.; formal analysis, S.A.; investigation, S.A. and T.A.B.; resources, S.A.; data curation, S.A. and T.K.; writing—original draft preparation, S.A.; writing—review and editing, M.N. and T.A.B.; visualization, S.K. and M.N.; supervision, T.A.B., M.N., and S.K.; project administration, S.K. and M.N. All authors have read and agreed to the published version of the manuscript.

**Funding:** This research received no external funding.

**Institutional Review Board Statement:** Not applicable.

**Informed Consent Statement:** Not applicable.

**Data Availability Statement:** The data that support the findings of this study are openly available in UC Irvine Machine Learning Repository [https://archive.ics.uci.edu/ml/datasets/Human+Activity+Recognition+Using+Smartphones, accessed on 10 December 2012] and [http://archive.ics.uci.edu/ml/datasets/mhealth+dataset, accessed on 7 December 2014].

**Conflicts of Interest:** The authors declare no conflict of interest.

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
