# Peer review of "Human Activity Classification Based on Angle Variance Analysis Utilizing the Poincare Plot"

_applsci, doi:10.3390/app11167230_

Round 1
Reviewer 1 Report
Previous questions were partially addressed.
# please add more details about the novelty in this research or the relevant research question : Corrected in the text
-Question: What is that correction and where is it in the text?
# If the hyperparameter tuning was done, please list the details : Corrected in the text page 8 line 254
- Does not answer the question.
it is recommended to add ROC curve : Corrected in the text page 10 in line 297
-OK
Reviewer 2 Report
Sufficiently updated to be published
Author Response
Thank you for you review.
Reviewer 3 Report
Dear authors, please answer a number of comments.
1. Update your bibliography.
2. Conclusions on the work should be made more detailed.
3. Part of the graphic material must be removed to the application.
4. Feasibility of Figures 1 and 2.
Round 2
Reviewer 1 Report
All questions are addressed.
Thank you.
This manuscript is a resubmission of an earlier submission. The following is a list of the peer review reports and author responses from that submission.
Round 1
Reviewer 1 Report
Even though human activity recognition using sensor data is a very important research area in the literature, but there are various shortcomings in the current state of the paper that requires attention before being published.
- The paper argues in the introduction that there has been limited research on feature engineering, which is not correct and numerous papers using conventional machine learning algorithms can be found in the literature.
- In addition, the introduction fails to clarify why there is a need to use such an alternative for activity detection. It would be useful to add relevant papers and the results to clarify what the shortcoming might have been.
- To my knowledge, the supervised deep-learning methods have shown high accuracy for human activity recognition and the paper fails to clarify why those studies are not enough that would justify the use of the Poincare Plot approach.
- The current state of the paper fails to clarify the data preparation and preprocessing pipeline. Please make sure to include details regarding the subjects and feature extraction.
There are few minor comments that might improve the state of the paper are listed below:
- Table 1. can be replaced with box plots that can illustrate the difference in distributions better.
- Table 2. includes too much information that makes it hard to follow. It might be better to replace that with a confusion matrix showing classification results for each activity and help compare them.
- In addition, please add a flowchart showing the pipeline and flow of the classification.
- The discussion needs to refer back to literature and compare results with numerous papers that are already published.
Reviewer 2 Report
In this work, the authors have proposed a single sensor-based activity classification method where Poincare plots have been introduced to analyze the variance of angle calculated from the raw accelerometer data. This is a very intriguing and interesting paper, with potential for large scale use.
Comments:
- Provide full forms of UCI, LSTM, etc.
- Provide more background information on HAR and MHealth Datasets, and clearly explain why they were selected.
- Explain in more detail the reasons for considering only Walking and Running data and choosing a chest-mounted sensor, in the case of the MHealth dataset
- Improvement opportunities / next steps towards practical applications should be discussed
Reviewer 3 Report
The authors present a study design that differentiates categories of activity using accelerometry. The stated contribution is the application of the Poincare plot to analyze data. While the use of Poincare is novel, the results do not go beyond multiple years of findings regarding activity classification with accelerometry. Overall, the presented work may be premature, and there are a lack of details regarding methodology. See below for detailed comments.
GENERAL
Generally, there have been extensive studies of accelerometry for activity recognition, with a number of examples of strong results. Therefore, the authors have not justified the need for the Poincare plot. Given Poincare is often used for analyses of nonlinear systems or chaotic data, so the need considering the data types (posture/activity) are not clear.
Authors do not justify the use of two separate datasets. The datasets are not combined, and essentially represent two separate studies/analyses. No contextualization is provided for use of the two datasets.
METHODOLOGY
Given the size of the HAR dataset (30 participants), authors should clarify the appropriateness of data analytic approaches.
The mHealth dataset is not described (e.g., number of participants, etc.)
Authors do not describe classification data. What percentage of data were used for training, testing, and validation?
Authors do not appear to have addressed drift/trend in raw data.
Authors have multiple examples of classification accuracies of 100%. Perfect classification may be a sign of overfitting study data.
IMAGES
Please provide units for Fig. 3 axes.
Reviewer 4 Report
#please add more details about the novelty in this research or the relevant research question
#If the hyperparameter tuning was done, please list the details
#it is recommended to add ROC curve